# Learning Sparse Structured Ensembles with SG-MCMC and Network Pruning

## Abstract

An ensemble of neural networks is known to be more robust and accurate than an individual network, however usually with linearly-increased cost in both training and testing. In this work, we propose a two-stage method to learn Sparse Structured Ensembles (SSEs) for neural networks. In the first stage, we run SG-MCMC with group sparse priors to draw an ensemble of samples from the posterior distribution of network parameters. In the second stage, we apply weight-pruning to each sampled network and then perform retraining over the remained connections. In this way of learning SSEs with SG-MCMC and pruning, we not only achieve high prediction accuracy since SG-MCMC enhances exploration of the model-parameter space, but also reduce memory and computation cost significantly in both training and testing of NN ensembles. This is thoroughly evaluated in the experiments of learning SSE ensembles of both FNNs and LSTMs. For example, in LSTM based language modeling (LM), we obtain 21% relative reduction in LM perplexity by learning a SSE of 4 large LSTM models, which has only 30% of model parameters and 70% of computations in total, as compared to the baseline large LSTM LM. To the best of our knowledge, this work represents the first methodology and empirical study of integrating SG-MCMC, group sparse prior and network pruning together for learning NN ensembles.

## 1 Introduction

Recently there are increasing interests in using ensembles of *Deep Neural Networks* (DNNs) (Ju et al. (2017); Huang et al. (2017)), which are known to be more robust and accurate than individual networks. An explanation stems from the fact that learning neural networks is an optimization problem with many local minima (Hansen & Salamon (1990)). Multiple models obtained from applying stochastic optimization, e.g. the widely used Stochastic Gradient Descent (SGD) and its variants, converge to different local minima and tend to make different errors. Due to this diversity, the collective prediction produced by an ensemble is less likely to be in error than individual predictions. The collective prediction is usually performed by averaging the predictions of the multiple neural networks.

On the other hand, the improved prediction accuracy of such model averaging can be understood from the principled perspective of Bayesian inference with Bayesian neural networks. Specifically, for each test point $\tilde{x}$, we consider the predictive distribution $P(\tilde{y}|\tilde{x}, \mathcal{D}) = \int P(\tilde{y}|\tilde{x}, \theta) P(\theta|\mathcal{D}) d\theta$, by integrating the model distribution $P(\tilde{y}|\tilde{x}, \theta)$ with the posterior distribution over the model parameters $P(\theta|\mathcal{D})$ given training data $\mathcal{D}$. The predictive distribution is then approximated by Monte Carlo integration $P(\tilde{y}|\tilde{x}, \mathcal{D}) \approx \frac{1}{M} \sum_{m=1}^{M} P(\tilde{y}|\tilde{x}, \theta^{(m)})$, where $\theta^{(m)} \sim P(\theta|\mathcal{D}), m = 1, \cdots, M$, are posterior samples of model parameters. It is well known that such Bayesian model averaging is more accurate in prediction and robust to over-fitting than point estimates of model parameters (Balan et al. (2015); Li et al. (2016); Gan et al. (2016)).

Despite the obvious advantages as seen from both perspectives, a practical problem that hinders the use of DNN ensembles in real-world tasks is that an ensemble requires too much computation in both training and testing. Traditionally, multiple neural networks are trained, e.g. with different random initialization of model parameters. Recent studies in (Loshchilov & Hutter (2016); Huang et al. (2017)) propose to learn an ensemble which consists of multiple snapshot models along the op-

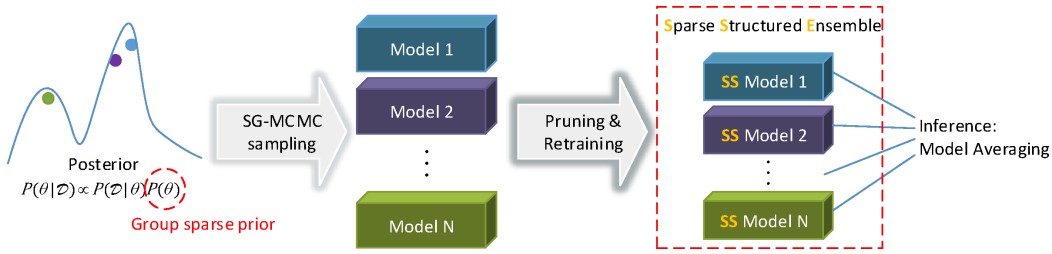

Figure 1: Overview of our two-stage method for learning SSEs.

timization path within a single training process, by leveraging a special cyclic learning rate schedule. This reduces the training cost, but the testing cost is still high.

In this paper we also aim at learning an ensemble within a single training process, but by leveraging the recent progress in Bayesian posterior sampling, namely the *Stochastic Gradient Markov Chain Monte Carlo* (SG-MCMC) algorithms. Moreover, we apply group sparse priors in training to enforce group-level sparsity on the network's connections. Subsequently we can further use model pruning to compress the networks so that the testing cost is reduced with no loss of accuracy.

Figure 1 presents a high-level overview of our two-stage method to learn Sparse Structured Ensembles (SSEs) for DNNs. Specifically, in the first stage, we run SG-MCMC with group sparse priors to draw an ensemble of samples from the posterior distribution of network parameters. In the second stage, we apply weight-pruning to each sampled network and then perform retraining over the remained connections as fine-tuning. In this way of learning SSEs with SG-MCMC and pruning, we reduce memory and computation cost significantly in both training and testing of NN ensembles, while maintaining high prediction accuracy. This is empirically verified in our experiments of learning SSE ensembles of both FNNs and LSTMs.

We evaluate the performance of the proposed method on two experiments with different types of neural networks. The first is an image classification experiment, which uses *Feed-forward Neural Networks* (FNNs) on the well-known MNIST dataset (Deng (2012)). Second, we experiment with the more challenging task of *Long Short-term Memory* (LSTM, Hochreiter & Schmidhuber (1997)) based language modeling, which is conducted on the Penn Treebank dataset (Marcus et al. (1993)). It is found that the proposed method works well across both tasks. For example, we obtain 12% relative reduction (from 78.4 to 68.6) in LM perplexity by learning a SSE of 4 large LSTM models, which has only 40% of model parameters and 90% of computations in total, as compared to the large LSTM LM in Zaremba et al. (2014). Furthermore, when the embedding weights of input and output are shared as in Inan et al. (2016), we obtain a perplexity of 62.1 (achieving 21% reduction from 78.4) by 4 large LSTMs with only 30% of model parameters and 70% of computations in total.

## 2 RELATED WORK

This work draws inspiration from three recent research findings, namely running SG-MCMC for efficient and scalable Bayesian posteriori sampling, applying group sparse priors to enforce network sparsity, and network pruning. To the best of our knowledge, this work represents the first methodology and empirical study of integrating these three techniques and demonstrates its usefulness to learning and using ensembles. In the following, more discussions are given on related studies.

**SG-MCMC sampling:** SG-MCMC represents a family of MCMC sampling algorithms developed in recent years, e.g. *Stochastic Gradient Langevin Dynamics* (SGLD) (Welling & Teh (2011), *Stochastic Gradient Hamiltonian Monte Carlo* (SGHMC) Chen et al. (2014), mainly for Bayesian learning from large scale datasets. SG-MCMC has the following favorable properties for learning ensembles. *(i)* SG-MCMC works by adding a scaled gradient noise during training, and thus enhances exploration of the model-parameter space. This is beneficial for finding diverse sample models for ensembles. *(ii)* Scalable and simple: the basic SG-MCMC algorithm, e.g. SGLD, is just a noisy *Stochastic Gradient Descent* (SGD), which means the same training cost as SGD on large

datasets. The effectiveness of applying SG-MCMC to Bayesian learning of RNNs is shown in Gan et al. (2016) but without considering model pruning to reduce the cost of model averaging.

**Sparse structure learning:** Group Lasso penalty (Yuan & Lin (2006)) has been widely used to regularize likelihood function to learn sparse structures. It was applied with SGD in (Wen et al. (2016); Alvarez & Salzmann (2016)) and in Wen et al. (2017) to learn structurally sparse DNNs and LSTMs respectively. But all focus on point estimates and are not in the context of learning ensembles. Group Lasso idea has been studied in Bayesian learning, which is known as applying group sparse priors (Marlin et al. (2009); Babacan et al. (2014)); but these previous works use variational method. Applying group sparse priors with SG-MCMC has not been explored.

**Model compression:** Model pruning and retraining (Han et al. (2015a), Hu et al. (2016)) has been studied to compress CNNs. Recently, Han et al. (2017) and Narang et al. (2017) apply model pruning to LSTM models for automatic speech recognition task. We use similar model pruning and retraining method in the experiments. We find that model averaging can enable the ensemble with heavily-pruned networks to be more robust in prediction.

**Learning ensembles:** Some efforts have been made to reduce the training and testing cost for ensembles. For reducing the training time cost of ensembles, a special cyclic learning rate schedule is developed in (Loshchilov & Hutter (2016); Huang et al. (2017)), which restarts the learning rate periodically to attempt to visit multiple local minima along its optimization path and saves snapshot models. In contrast to relying on such empirical setting of the learning rate to explore model space, theoretical consistency properties of SG-MCMC methods in posterior sampling have been established (Teh et al. (2016)). For reducing the testing time cost of ensembles, Hinton et al. (2015) and Balan et al. (2015) distill the knowledge of an ensemble into a single model, but still require large training cost.

## 3 LEARNING ENSEMBLES WITH SG-MCMC AND NETWORK PRUNING

We consider the classification problem under Bayesian inference framework. Given training data $\mathcal{D} \triangleq \{(x_i, y_i)\}_{i=1}^N$ with input feature $x_i \in \mathbb{R}^D$ and class label $y_i \in \mathcal{Y}$, where $\mathcal{Y}$ is the set of classes. We view a neural network as a conditional probabilistic model $P(y_i|x_i, \theta)$. Denote the network parameters by $\theta$, with $P(\theta)$ a prior distribution. We compute the posterior distribution over the model parameters, $P(\theta|\mathcal{D}) \propto P(\theta) \prod_{i=1}^N P(y_i|x_i, \theta)$. For testing, given a test input $\tilde{x}$, the Bayesian predictive distribution for its label $\tilde{y}$ is given by $P(\tilde{y}|\tilde{x}, \mathcal{D}) = \mathbb{E}_{P(\theta|\mathcal{D})}[P(\tilde{y}|\tilde{x}, \theta)]$, which can be viewed as model averaging across parameters with distribution $P(\theta|\mathcal{D})$. However, the integration over the posterior is analytically intractable for deep neural networks (DNNs). Thus it is approximated by Monte Carlo integration as in the following:

$$P(\tilde{y}|\tilde{x}, \mathcal{D}) \approx \frac{1}{M} \sum_{m=1}^M P(\tilde{y}|\tilde{x}, \theta^{(m)}), \quad \theta^{(m)} \sim P(\theta|\mathcal{D}) \tag{1}$$

where $\{\theta^{(m)}\}_{m=1}^M$ is a set of posterior samples drawn from $P(\theta|\mathcal{D})$, e.g. by the popular Markov Chain Monte Carlo (MCMC) methods. Traditional MCMC methods either have low-efficiency for high dimensional sampling or scale poorly with dataset. Fortunately, the recently developed SG-MCMC methods work on stochastic gradients over small mini-batches, which alleviate these problems and can be applied for posterior sampling for DNNs.

### 3.1 SAMPLING VIA STOCHASTIC GRADIENT LANGEVIN DYNAMICS

Specifically, we choose the simplest and most widely used SG-MCMC algorithm - *Stochastic Gradient Langevin Dynamics* (SGLD) (Welling & Teh (2011)) as the sampling method in our first stage of learning ensembles. Extension by using other high-order SG-MCMC algorithms is straightforward. SGLD calculates a stochastic gradient of negative log posterior based on $S_t$, small mini-batch of training data:

$$\widetilde{g}_t \triangleq \nabla_\theta \widetilde{U}_t(\theta) = -\frac{N}{|S_t|} \sum_{i \in S_t} \nabla_\theta \log P(y_i|x_i, \theta) - \nabla_\theta \log P(\theta) \tag{2}$$

where $U(\theta) \triangleq -\log P(\mathcal{D}|\theta) - \log P(\theta)$ is known as the potential energy in SG-MCMC sampling and $\widetilde{U}_t(\theta)$ is its approxmation over the $t$-th mini-batch. The updating rule of SGLD is as simple as SGD with an additional Gaussian noise $\xi \sim \mathcal{N}(0, \epsilon_t \mathrm{I})$ as following:

$$\theta_{t+1} = \theta_t - \frac{\epsilon_t}{2}\widetilde{g}_t + \xi \tag{3}$$

where $\epsilon_t$ is the learning rate or step size. By using gradient information and stochastic mini-batch updating, SGLD overcomes the problems in traditional MCMC methods and thus leads to efficient posterior sampling.

In the following, we provide three discussions about applying SGLD to learning ensembles. First, note that SGLD is proposed with the use of annealing learning rates since SGLD does not have a Metropolis-Hastings correction step; discretization error goes to zero only when learning rates annealed to zero. In spite of that, some studies suggest to use constant learning rates in practice (Sato & Nakagawa (2014), Chaudhari et al. (2016)), which is found to give better mixing rate and make more extensive exploration of parameter space. This is also compatible with our aim of learning a good ensemble, since we want to collect diverse models. We test both annealing and constant learning rates in our experiments and find that using constant learning rates performs better, as expected. Hence, we only report the results of using constants learning rate in this work.

Second, we need to consider how to sample $\theta$ from the parameter updating sequence $\{\theta_t\}_{t=1}^{T}$, where $T$ is the total number of iterations. Firstly, a burn-in process is desired. Secondly, a thinned collection of samples $\{\theta_{\frac{kT}{M}}\}_{k=1}^{M}$ performs better than other strategies like backward collection $\{\theta_t\}_{t=T-M+1}^{T}$, since there are lower correlations between samples. Our preliminary results as well as the results from Gan et al. (2016) both hold for that, so we take thinned collection as the default setting in this work.

Finally, we need to consider how long to run the sampling algorithm and how many models are used for model averaging. The fixed-scale additional noise in SGLD generally reduces overfitting, thus longer running can be allowed in order to better explore the parameter space. As shown by the empirical result in Fig. 3(b), the SGLD learning method indeed can improve performance by averaging more models than other traditional methods of learning ensembles.

### 3.2 PRUNING AND RETRAINING

After all the models are collected, we come to the second stage of learning DNN ensembles - network pruning and retraining. We use a simple pruning rule, i.e. finding the network connections whose weights are below certain threshold and removing them away, as did in (Han et al. (2015b)). The threshold is determined by the configured overall sparsity or pruning ratio, e.g. 90%, after sorting the weights by their absolute values.

Once the network is pruned, the posterior changes from $P(\theta|\mathcal{D})$ to the reduced posterior $Q^{(m)}(\phi^{(m)}|\mathcal{D})$, where $m$ is the index of the pruned network. Retraining is then performed for each pruned network:

$$\hat{\phi}^{(m)} = \arg\max_{\phi^{(m)}} \log Q^{(m)}(\phi^{(m)}|\mathcal{D}), \quad m = 1, 2, \ldots, M \tag{4}$$

We thus obtain an ensemble of networks $\{\hat{\phi}^{(m)}\}_{m=1}^{M}$, which are in fact maximum a posterior (MAP) estimates under the reduced posteriors.

The effect of pruning is to reduce the model size as well as the computation cost. Interestingly, it is found in our experiments that retraining of the sampled models, whether being pruned or not, significantly improve the performance of the ensemble. There are two justifications for the retraining phase. First, theoretically (namely with infinite samples), model averaging according to Equ. (1) does not need retraining. However, the actual number of samples used in practice is rather small for computational efficiency. So retraining essentially compensates for the limited size of samples for model averaging. Second, if we denote by $\bar{\phi}^{(m)}$ the network obtained just after pruning but before retraining, it can be seen that the MAP estimate $\hat{\phi}^{(m)}$ is more likely than $\bar{\phi}^{(m)}$ under the reduced posterior. Note that the probability of $\bar{\phi}^{(m)}$ under the reduced posterior is close to the probability of $\bar{\phi}^{(m)}$ under the original posterior, since we only prune small network weights. So retraining

increases the posteriori probabilities of the networks in the ensemble and hopefully improves the prediction performance of the networks in the ensemble.

## 4 SPARSE STRUCTURED ENSEMBLES

The main computation for training or testing a DNN is the large amount of matrix calculations, which are commonly accelerated by using a GPU hardware. However, a randomly pruned network is not friendly for GPUs to handle, since the randomly positioned zeros in the weight matrices still require *floating-point operations* (FLOPs) without special treatment. In our *Sparse Structured Ensembles* (SSEs), we take this into consideration and aim at learning structures for reducing FLOPs in the sense of matrix calculations.

### 4.1 GROUP SPARSE PRIOR

In optimization, a regularization term is often used as a penalty to the objective function to do trade-off between minimizing a loss function and choosing a desirable model with certain constraints. The group Lasso regularization Yuan & Lin (2006) proposes to do feature selection in group level, which means keeping or removing all the parameters in a group simultaneously to achieve structured sparsity corresponding to grouping strategy. It can be formulated as:

$$R(\theta) = \lambda \sum_{g=1}^{G} \sqrt{dim(\theta_g)} \left\| \theta_g \right\|_2 \tag{5}$$

where $\theta_g$ is a group of weights in $\theta$, $G$ is the number of groups, $dim(\theta_g)$ denotes the number of weights in $\theta_g$ and $\| \cdot \|_2$ denotes the $l_2$ norm. The term $\sqrt{dim(\theta_g)}$ ensures that each group gets regularized uniformly corresponding to its dimension. The coefficient $\lambda$, called GSP strength coefficient, is a hyperparameter to do trade off between gaining group sparsity and minimizing the loss function. While in training, the gradient of each component can be calculated by

$$\frac{\partial \sqrt{dim(\theta_g)} \left\| \theta_g \right\|_2}{\partial \theta_g} = \sqrt{dim(\theta_g)} \frac{\theta_g}{\|\theta_g\|_2} \tag{6}$$

A small constant could be added to $\|\theta_g\|_2$ in order to avoid the denominator being zero. In our experiments, we find $\|\theta_g\|_2$ fluctuate near zero and thus do not add the constant.

In Bayesian inference framework, the regularization term corresponds to the negative log prior term $-\log P(\theta)$ in the potential energy $U(\theta)$, thus the group Lasso regularization can be converted into a specific prior as follows:

$$P(\theta) = \frac{1}{Z} \exp(-R(\theta)) = \frac{1}{Z} \exp(-\lambda \sum_{g=1}^{G} \sqrt{dim(\theta_g)} \left\| \theta_g \right\|_2) \tag{7}$$

where $Z$ is a normalization constant. The gradient term $-\nabla_\theta \log P(\theta)$ in Equ. (2) for SGLD parameter updating can be directly calculated via Equ. (7), without the use of complex hierarchical decomposition form for the prior as the variational methods do (Marlin et al. (2009); Babacan et al. (2014)). We call it a *Group Sparse Prior* (GSP) as named in Marlin et al. (2009).

### 4.2 GROUPING STRATEGIES

To learn sparse structured networks for our SSE, it is necessary to specify grouping strategy according to the characteristics of different types of neural networks. In this paper, we show how to learn SSE for both FNN and LSTM. Their grouping strategies are described separately in the following.

**Feed-forward Neural Network:** For FNN, we group all the outgoing connections from a single neuron (input or hidden) together following Scardapane et al. (2017). Since FNN's simple hierarchical structure, if a neuron's outputs are all zeros, it makes no contribution to the next layer and can be removed. This leads to node pruning instead of random connection pruning, which reduces the rows and columns of weight matrices between layers, thus leading to lower matrix-level FLOPs as

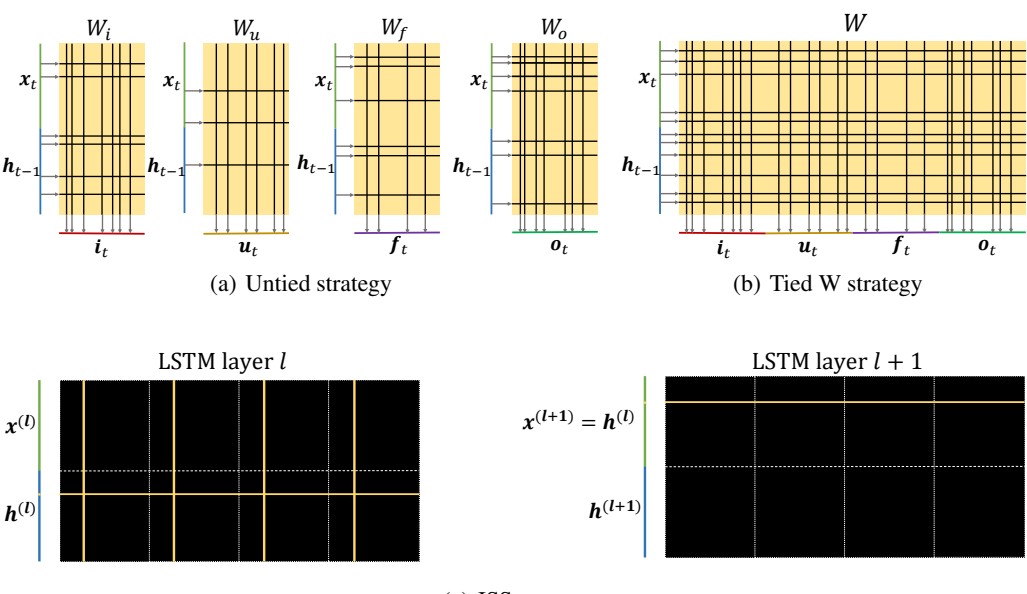

Figure 2: Illustration of different grouping strategies for LSTMs. In figure (a) and (b), the black lines in each weight matrix represent non-zero elements and the yellow areas are rows and columns with all zeros, as a result of group selection enforced by GSP. The horizontal arrows indicate the input and hidden units used in calculations, and the vertical arrows point to those dimensions of a gate to be updated. Thus it is enough to do calculations by the reduced matrix formed by these black rows and columns in the figure instead of the whole matrix. In figure (c), the dash white lines separate each weight matrix of tied W matrix. Yellow lines indicate the weights associated with a certain hidden unit in LSTM layer $l$, which are removed simultaneously to reduce the hidden size of $h^{(l)}$.

expected. We can also group the incoming connections of a neuron, but the neurons with no incoming weights are still required to shift their biases to the next layer, which is a bit more complex than the above strategy we choose.

**Long Short-term Memory:** The case is not that simple for LSTM, since the input and hidden units are used four times when calculating the input gate, forget gate, cell updates and output gate as follows:

$$
\begin{aligned}
\boldsymbol{f}_t &= \sigma([\boldsymbol{x}_t, \boldsymbol{h}_{t-1}]W_f + \boldsymbol{b}_f) & \boldsymbol{i}_t &= \sigma([\boldsymbol{x}_t, \boldsymbol{h}_{t-1}]W_i + \boldsymbol{b}_i) \\
\boldsymbol{u}_t &= \tanh([\boldsymbol{x}_t, \boldsymbol{h}_{t-1}]W_u + \boldsymbol{b}_c) & \boldsymbol{o}_t &= \sigma([\boldsymbol{x}_t, \boldsymbol{h}_{t-1}]W_o + \boldsymbol{b}_o) \\
\boldsymbol{c}_t &= \boldsymbol{f}_t \odot \boldsymbol{c}_{t-1} + \boldsymbol{i}_t \odot \boldsymbol{u}_t & \boldsymbol{h}_t &= \boldsymbol{o}_t \odot \tanh(\boldsymbol{c}_t)
\end{aligned}
\tag{8}
$$

where all the vectors are row vectors, $\sigma(\cdot)$ is the sigmoid function, $[\cdot, \cdot]$ denotes concatenating horizontally and $\odot$ is element-wise multiplication. Removing an input or hidden unit is difficult for LSTM since every unit affects all the updating steps. However, note that the weight matrix between units and each gate is fully-connected, it is still beneficial to reduce the matrix size by removing a row or column. Specifically, we keep two index lists during pruning to record the remained rows and columns for each weight matrix. When doing computations, we just use partial units to update partial dimensions of the gates according to the index lists. This is flexible for different units to provide updating for different gate dimensions.

Thus, our first grouping strategy is to group each row and each column for the four weight matrices separately in Equ. (8). Note that the group sparse prior generally selects or removes a certain group, it is allowed to make groups overlapped for reducing matrix size. We consider this untied strategy since the most basic implementation of LSTM cell conducts calculation as in Equ. (8). Alternatively,

the LSTM updating formulas can be written as in Gal & Ghahramani (2016):

$$
\begin{pmatrix} \boldsymbol{i}_t \\ \boldsymbol{u}_t \\ \boldsymbol{f}_t \\ \boldsymbol{o}_t \end{pmatrix} = \begin{pmatrix} \sigma \\ \tanh \\ \sigma \\ \sigma \end{pmatrix} \left( \begin{pmatrix} \boldsymbol{x}_t \\ \boldsymbol{h}_{t-1} \end{pmatrix} \cdot W \right)
\tag{9}
$$

where $W$ is a matrix of dimension $2n$ by $4n$ ($n$ being the unit number for a hidden state), which is the horizontally concatenation of the four weight matrices in Equ. (8). Since acceleration has been reported by concatenating matrix (Appleyard et al. (2016)), we also try to group each row and column of $W$ as a second grouping strategy. This strategy is simpler since only two index lists are kept instead of eight, and we call it *tied W* strategy.

In a concurrent work of learning structurally sparse LSTMs Wen et al. (2017) using SGD, a grouping strategy, called *Intrinsic Sparse Structures* (ISS), is proposed to reduce the hidden size by grouping all the weights associated with a certain hidden unit together and removing them simultaneously. LSTMs learned by ISS can be reconstructed easily with the pruned smaller hidden size, without the need to keep original model size and index lists. However, the embedding size is not reduced in (Wen et al. (2017)), which leads to high cost in computing the input for the 1st LSTM layer. To overcome this, there are two schemes. (a) Each column of the input embedding matrix is grouped to further reduce the input size of the 1st LSTM layer; (b) The weights from the embedding layer and the softmax layer are shared, as proposed in Inan et al. (2016), thus the embedding size is the same as hidden size of the last LSTM layer. An illustration of these strategies are shown in Fig. 2(c).

## 5 EXPERIMENTS

In our experiments, we implemented the proposed method in TensorFlow, and present the results in two parts: *(i)* learn SSE of FNNs for image classification task on MNIST; *(ii)* learn SSE of LSTM models, which is more challenging, for word-level language modeling task on Penn TreeBank corpus.

The sparsity of a network in this paper means the percentage of the pruned weights from the total parameters, FLOPs for a matrix $W$ is calculated as the size of the smallest sub-matrix formed by such rows and columns in $W$ that contain all non-zero elements in $W$, and FLOPs for a network is the sum of FLOPs for all its weight matrices. The parameters and FLOPs presented in the following tables are the total size considering all the models in an ensemble, unless otherwise indicated. PR, GSP and SSE denotes Pruning and Retraining, Group Sparse Prior and Sparse Structured Ensemble respectively.

### 5.1 CLASSIFICATION ON MNIST

First we use our method on FNNs for classification on the well-known MNIST dataset. We choose a commonly used network structure of 2 hidden layers with 300 and 100 hidden neurons respectively and ReLU activations, denoted as FNN-784-300-100-10. We run our experiments without any additional tricks such as dropout, batch normalization etc. Such basic setting allows easy reproduction of the results. All the results reported in table 1 are averaged results from 10 independent runs. The detailed model structure information for one arbitrary model taken is shown in table 2.

The baseline FNN-784-300-100-10 is trained by Stochastic Gradient Descent (SGD). Specifically it is trained for 100 epochs with an annealing learning rate of 0.5 which decays by a factor of 2 every 10 epoch. The baseline obtains 1.66% test error rate, and an ensemble of 18 independently trained FNN-784-300-100-10 networks decrease the error to 1.49%.

In the first group of experiments with SGLD learning, we use Laplace priors, similar to adding L1 regularization. We train also for 100 epochs but with a constant learning rate of 0.5. Network samples are collected every 5 epoch after a 10 epoch burn in. The ensemble learned by SGLD gives 1.53% test error, which is slightly worse than the independently trained ensemble, since each sample drawn by constant-step-size SGLD sampling is not as accurate as the sample trained through SGD optimization. Adding L1 could enforce sparse structure learning and allows us to prune the network weights. After pruning, we retrain each network in the ensemble for 20 epochs with a small learning rate of 0.01 which decay by factor 1.15 every epoch. The resulting ensembles are denoted

Table 1: MNIST results of various models based on FNN-784-300-100-10. The number of parameters and FLOPs are shown as multiples of the baseline FNN trained by SGD, the specifics of which are shown in parentheses. PR: Pruning and Retraining, GSP: Group Sparse Prior.

| Method | Model | Parameters | FLOPs | Test Error (%) |
|---|---|---|---|---|
| SGD (baseline) | 1 model | 1 (266K) | 1 (532K) | 1.66 |
| SGD | 18 models | 18× | 18× | 1.49 |
| SGLD | 18 models | 18× | 18× | 1.53 |
| SGLD+L1+PR | 18 models, 90% sparsity | 1.8× | 3.4× | 1.26 |
| SGLD+L1+PR | 18 models, 96% sparsity | 0.7× | 3.0× | 1.39 |
| SGLD+GSP+PR | 18 models, 90% sparsity | 1.8× | 2.5× | **1.26** |
| SGLD+GSP+PR | 18 models, 96% sparsity | **0.7×** | **2.2×** | 1.29 |

Table 2: Detailed structure information of various FNN ensembles based on FNN-784-300-100-10 for MNIST.

| Model | Sparsity | Parameters | Network structure | FLOPs |
|---|---|---|---|---|
| SGLD 18 models | - | 266K | 784-300-100-10 | 532K |
| SGLD+GSP+PR 18 models | 90% | 27K | 380-128-24-10 | 75K (14%) |
| SGLD+GSP+PR 18 models | 96% | 11K | 364-82-22-10 | 64K (12%) |

by SGLD+L1+PR in Table 1. For these ensembles, the highest sparsity without losing accuracy is 90%. When 96% of the parameters are pruned, the performance is worsened obviously.

In the second group of experiments with SGLD learning, our new method, SGLD with group sparse prior (GSP) is applied. The resulting ensembles are denoted by SGLD+GSP+PR in Table 1. When compared to SGLD+L1+PR, the new method achieves larger sparsity (up to 96%) and FLOP reduction without losing accuracy, presumably because applying GSP forces the pruned connections to be aligned, thus removes more neurons. When compared to the baseline FNN, the SSE of 18 networks learned by the new method decreases test error from 1.66% to 1.29% with 70% of parameters and 2.2× computational cost.

## 5.2 Language Modeling

Next, we experiment with the more challenging task of learning ensembles of LSTMs, which represent a widely used type of recurrent neural networks (RNNs) for sequence learning tasks. Specifically, we study the task of LSTM-based language modeling (LM), which basically is to predict the next word given previous words. The prediction performance is measured by perplexity (PPL), which is defined as the exponential of negative log-probability per token. A popular LM benchmarking dataset - Penn TreeBank (PTB) corpus (Marcus et al. (1993)) is used, with a vocabulary of 10K words and 929K/73K/10K words in training, development and test sets respectively.

We use Zaremba et al. (2014) as the baseline and follow their LSTM architectures to make comparable results. We test different methods on the medium (2 layers with 650 hidden units each) and large (2 layers with 1500 hidden units each) LSTM models as used in Zaremba et al. (2014). The dimension of word embedding as input is the same as the size of hidden units. All the models are trained with the dropout technique introduced in Zaremba et al. (2014). The experiments without GSP just follow their dropout keep ratio which are 0.5 and 0.35 for medium and large model respectively. It is found in our experiments that when applying GSP, a higher dropout keep ratio is desired, which are 0.7 for medium and 0.5 for large model. This is presumably because that both GSP and dropout are some form of regularization and regularizing too much will lead to underfitting. For both untied and tied cases of LSTM weight matrices, the GSP strength coefficients are $\lambda = 4.2 \times 10^{-5}$ and $\lambda = 2.3 \times 10^{-5}$ respectively for medium and large model; for the ISS case, the GSP strength coefficients are $\lambda = 3.0 \times 10^{-5}$ and $\lambda = 1.5 \times 10^{-5}$ respectively. For both medium and large models, the

Table 3: Model ablations for the new method SGLD+GSP+PR based on the medium LSTM LMs over PTB. The column of single model denotes the lowest PPL obtained by a single model in the ensemble. The number of parameters and FLOPs are shown as multiples of the baseline medium LSTM trained by SGD, the specifics of which are shown in parentheses. The grouping strategy is the untied weight strategy by default, unless specified in parentheses. Tied W denotes tied weight strategy and ISS denotes *Intrinsic Sparse Structures* as in Wen et al. (2017)

| Method | Model | Parameters | FLOPs | Single model | | Ensemble | |
|---|---|---|---|---|---|---|---|
| | | | | Dev. | Test | Dev. | Test |
| SGD (Zaremba, 2014) | 1 | 1 (19.8M) | 1 (26.5M) | 86.2 | 82.1 | - | - |
| SGD (Zaremba, 2014) | 10 | 10× | 10× | - | - | 75.2 | 72.0 |
| SGLD | 10 | 10× | 10× | 87.0 | 83.7 | 80.5 | 78.9 |
| SGLD+PR | 10 | 1× | 10× | 103.8 | 100.2 | 91.1 | 89.4 |
| SGLD+GSP | 10 | 10× | 10× | 98.8 | 97.0 | 88.0 | 86.9 |
| SGLD+GSP+R | 10 | 10× | 10× | 80.0 | 76.3 | 70.8 | 69.1 |
| SGLD+GSP+P | 10 | 1× | 4× | 103.8 | 101.9 | 96.1 | 94.7 |
| SGLD+GSP+PR | 10 | 1× | 4× | 79.8 | 76.6 | 71.5 | 69.5 |
| SGLD+GSP+PR (tied W) | 10 | 1× | 4× | 79.7 | 76.6 | 70.9 | 69.2 |
| SGLD+GSP+PR (ISS) | 10 | 1× | 3× | 80.9 | 77.4 | 71.8 | 69.9 |

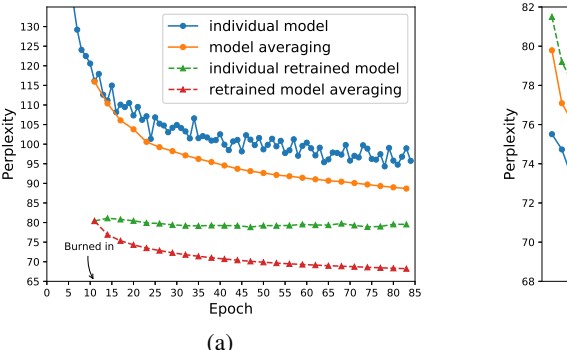
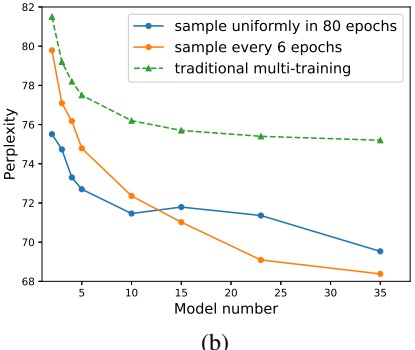

(a)                                             (b)

Figure 3: (a) The PPL curves along the learning of LSTM ensemble by SGLD+GSP+PR over the PTB development set. It is worthwhile to note that as the training proceeds, more models are averaged, which consistently improves the PPLs. (b) The PPLs over development set v.s. the number of models in the LSTM ensembles by SGLD+GSP+PR.

learning rates are fixed to 1.5 and 1.0 respectively for SGLD training, and decay by a factor of 1.25 for retraining by SGD. All the hyperparameter settings above are found empirically via grid search on the validation set.

The results are organized into four parts. (1) For ablation analysis, we study the contribution of each component in the new method SGLD+GSP+PR through a series of comparison experiments, as shown in Table 3. (2) We show the effects of the number of model samples and sampling strategies for the method SGLD+GSP+PR, as given in Fig. 3. (3) We display in Fig. 4 the sparse structures, obtained from applying the method SGLD+GSP+PR. (4) Main results are summarized and compared in Table 5.

Table 3 lists the model ablation results of SGLD training with different combinations of pruning, retraining and GSP (untied grouping strategy as default). It is found that applying PR or GSP alone lead to worse performance than vanilla SGLD for learning LSTM ensembles. When they are applied together, GSP forces the network to learn group sparse structures which are highly robust to pruning, thus leading to a better result. With GSP, applying pruning only leads to negligible loss of perfor-

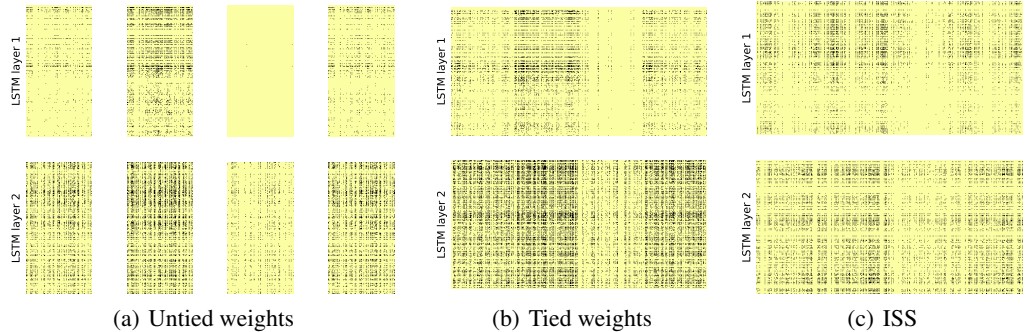

|  (a) Untied weights | (b) Tied weights | (c) ISS |

Figure 4: The sparse structure patterns of the weight matrices from a sample LSTM model trained by applying SGLD with GSP. Yellow areas are all zeros while black dots stand for non-zero weights. The patterns are plotted with a sub-sampling by a factor of 5.

Table 4: Detailed structure information for various large LSTMs. The FLOPs of LSTM layer 1, L-STM layer 2 and softmax layer are shown in three columns respectively. The embedding layer is not listed here since it is a table lookup process instead of matrix calculation. The results of our method are the statistics from a single model sample from the ensemble trained by SGLD+GSP+PR. The size of the reduced LSTM model learned by our method with ISS is 365, 311 and 420 for embedding input, 1st LSTM layer and 2nd LSTM layer respectively. For shared embeddings (denoted as SE), the layer sizes are 456, 352 and 456 respectively.

| Method | Sparsity | Parameters | FLOPs | | | |
| --- | --- | --- | --- | --- | --- | --- |
| | | | LSTM1 | LSTM2 | Softmax | Total |
| SGD (Zaremba, 2014) | - | 66M | 36M | 36M | 30M | 102M |
| ISS (Wen, 2017) | - | 25.2M | 5.7M | 3.9M | 10.7M | 20.4M (20%) |
| SGLD+GSP+PR | 90% | 6.6M | 4.3M | 6.1M | 12.2M | 22.7M (22%) |
| SGLD+GSP+PR (tied W) | 90% | 6.6M | 4.9M | 6.1M | 12.4M | 23.5M (23%) |
| SGLD+GSP+PR (ISS) | 90% | 6.6M | 1.7M | 2.5M | 8.4M | 12.6M (12%) |
| SGLD+GSP+PR+SE (ISS) | 90% | 5.1M | 2.3M | 2.9M | 9.1M | 14.4M (14%) |

mance but greatly reduces the model size, as can been seen from comparing SGLD+GSP+PR with SGLD+GSP+R; pruning without retraining produces inferior result. The three grouping strategies perform close to each other. Remarkably, compared to the medium LSTM ensemble obtained by multiple training, the ensemble learned by the new method SGLD+GSP+PR reduces the PPL from 72.0 to 69.5, and with only 10% parameters and 40% FLOPs in total. SGLD indeed provides a good approach to finding diverse sample models for ensembles.

Fig.3(a) presents the PPL curves along the learning of LSTM ensemble by the new method S-GLD+GSP+PR over the development set. It clearly shows the performance gains brought by model averaging and PR. The relationship between the performance of an ensemble and the number of models in an ensemble is examined in Fig.3(b), together with a comparison between different sampling strategies. We test the performances of different ensembles on the PTB development set, each consisting of 2 to 35 medium LSTMs. The blue curve shows the result of running SGLD+GSP+PR for 80 epochs and sampling uniformly after a 10-epoch burn in process, *e.g.* sampling every 2 epoch to collect 35 models and sampling every 35 epoch to get 3 models. The orange curve is obtained by sampling every 6 epochs, which means that the more model collected the longer run of SGLD. It can be seen from comparing the two curves that it is better to sample with larger interval with relatively small number of models. It is also clear from Fig.3(b) that the traditional ensemble learning method by SGD training of multiple models is inferior to the SGLD learning method.

Table 5: Comparison of various models based on LSTMs on PTB dataset. The number of parameters and FLOPs are shown as multiples of the baseline medium LSTM trained by SGD, the specifics of which are shown in parentheses. The bold line denotes the best result obtained with shared embeddings (denote as SE).

| Method | Model | Parameters | FLOPs | Dev. | Test |
|---|---|---|---|---|---|
| SGD (Zaremba, 2014) | 1 large | 1(66M) | 1(102M) | 82.2 | 78.4 |
| SGD (Zaremba, 2014) | 38 large | 38× | 38× | 71.9 | 68.7 |
| VD (Gal, 2016) | 10 large | 10× | - | - | 68.7 |
| VD-LSTM+SE+AL (Inan, 2016) | individual | 51M | - | 71.1 | 68.5 |
| AWD-LSTM (Merity, 2017) | individual | 24M | - | 60.0 | 57.3 |
| AWD-LSTM-MoS (Yang, 2017) | individual | 22M | - | 56.5 | 54.4 |
| SGD+GSP+PR | 1 large | 0.1× | 0.2× | 81.9 | 77.8 |
| SGD+GSP+PR | 4 large | 0.4× | 0.9× | 69.9 | 66.7 |
| SGLD+GSP+PR | 20 large | 2.0× | 4.5× | 68.6 | 66.4 |
| SGLD+GSP+PR | 4 large | 0.4× | 0.9× | 70.9 | 68.7 |
| SGLD+GSP+PR+SE (ISS) | 4 large | **0.3× (20.4M)** | **0.7 ×** | **64.4** | **62.1** |

Fig. 4 shows the sparse structured patterns of the weight matrices from a single LSTM model sample in the ensemble trained by applying SGLD with GSP, separately for three different grouping strategies of weight matrices. Table 4 shows the FLOPs for each layer for these LSTM model samples. Note that word embedding is not included for FLOP calculation, since it is a table lookup process instead of matrix calculation in practice, but we still prune it to reduce model size. Models learned by ISS have desirable homogeneously-sparse structures and thus fewer FLOPs.

Comparison of various models based on LSTMs on PTB dataset are summarized in Table 5. We investigate to use small number of models to achieve trade off between cost and performance. An attractive model is the SSE of 4 large LSTMs, which only requires 40% of parameters and 90% of computational cost in total, compared to the baseline large LSTM Zaremba et al. (2014), but decrease the perplexity from 78.4 to 68.7. This result is also better than those obtained by Zaremba et al. (2014) (38 independently trained large LSTMs) and Gal & Ghahramani (2016) (10 independently trained large LMs with costly MC dropout for testing), not only in terms of PPL reduction but also in term of reducing memory and computing costs.

As suggested by a referee, we compare SGD (1 model)+GSP+PR with SGLD (ensemble)+GSP+PR. SGD+GSP+PR represents the SGD training with group sparse prior and model pruning/retraining. SGD (1 model)+GSP+PR can reduce the model size but the PPL is much worse than the ensemble, which clearly shows the improvement provided by the ensemble. Additionally, we compare SGLD (4 models)+GSP+PR with SGD (4 models)+GSP+PR, namely the classic ensemble training method by multiple independent runs with different initializations. The two ensembles achieve close PPLs. However, SGD ensemble learning requires $30 \times 4$ epochs training and $15 \times 4$ epochs retraining, SGLD ensemble learning takes 80 epochs training plus $15 \times 4$ epochs retraining, which reduces about $30\%$[1] training time.

Note that a number of better model architectures (Inan et al. (2016); Merity et al. (2017); Yang et al. (2017)) have emerged over the baseline large LSTM LM, which we used as a baseline. Our SGLD+GSP+PR in principle can be applied to those new model architectures. As an example, we apply SGLD+GSP+PR to models that share input and output embeddings Inan et al. (2016). With shared embeddings, we further reduce the perplexity to 62.1 by using the SSE of 4 large LSTMs, which can be regarded as an ensemble version of Inan et al. (2016) without variational dropout (VD) and augmented loss (AL). As a side note, without shared embeddings, the lowest perplexity achieved is 66.4 by the SSE of 20 large LSTMs, which is also among the top models obtained with standard LSTMs to the best of our knowledge.

---

[1]Retraining operates on pruned models, and reduces the time cost by $50\%$. So the total reduction of training time is about $(80 + 15 * 4 * 0.5)/(30 * 4 + 15 * 4 * 0.5) = 0.73$.

# 6 CONCLUSION AND FUTURE WORKS

In this work, we propose a novel method of learning NN ensembles efficiently and cost-friendly by integrating three mutually enhanced techniques: SG-MCMC sampling, group sparse prior and network pruning. The resulting SGLD+GSP+PR method is easy to implement, yet surprisingly effective. This is thoroughly evaluated in the experiments of learning SSE ensembles of both FNNs and LSTMs. The Sparse Structured Ensembles (SSEs) learned by our method gain better prediction performance with reduced training and test cost when compared to traditional methods of learning NN ensembles. Moreover, by proper controlling the number of models used in the ensemble, the method can also be used to produce SSE, which outperforms baseline NN significantly without increasing the model size and computation cost.

Some interesting future works: (1) interleaving model sampling and model pruning; (2) application of this new method, as a new powerful tool of learning ensembles, to more tasks.

## ACKNOWLEDGMENTS

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
