# OpenReview forum: "Learning Sparse Structured Ensembles with SG-MCMC and Network Pruning"
_ICLR.cc/2018/Conference — Reject_

### Official Review · AnonReviewer1 · 2017-11-27
**Justification for the proposed algorithm is weak + weak experiments.**

**Rating:** 4
**Confidence:** 4

**Review:**

The authors propose a procedure to generate an ensemble of sparse structured models. To do this, the authors propose to (1) sample models using SG-MCMC with group sparse prior, (2) prune hidden units with small weights, (3) and retrain weights by optimizing each pruned model. The ensemble is applied to MNIST classification and language modelling on PTB dataset.

I have two major concerns on the paper. First, the proposed procedure is quite empirically designed. So, it is difficult to understand why it works well in some problems. Particularly. the justification on the retraining phase is weak. It seems more like to use SG-MCMC to *initialize* models which will then be *optimized* to find MAP with the sparse-model constraints. The second problem is about the baselines in the MNIST experiments. The FNN-300-100 model without dropout, batch-norm, etc. seems unreasonably weak baseline. So, the results on Table 1 on this small network is not much informative practically. Lastly, I also found a significant effort is also desired to improve the writing.

The following reference also needs to be discussed in the context of using SG-MCMC in RNN.
- "Scalable Bayesian Learning of Recurrent Neural Networks for Language Modeling", Zhe Gan*, Chunyuan Li*, Changyou Chen, Yunchen Pu, Qinliang Su, Lawrence Carin

---

> ### Author Response · Authors · 2017-12-22
> **Response to Reviewer1**
>
> Thank you very much for reviewing the paper.
>
> > Particularly. the justification on the retraining phase is weak.
>
> Thanks for your note. As stated in the end of Section 3.2, there are two justifications for the retraining phase: First, theoretically (namely with infinite samples), model averaging does not need retraining. However, the actual number of samples used in practice is rather small for computational efficiency. So retraining essentially compensates for the limited size of samples for model averaging. Second, the MAP estimate is more likely than the network obtained just after pruning but before retraining. Retraining increases the posteriori probabilities of the networks in the ensemble and hopefully improves the prediction performance of the networks in the ensemble.
>
> > The second problem is about the baselines in the MNIST experiments. The FNN-300-100 model without dropout, batch-norm, etc. seems unreasonably weak baseline. So, the results on Table 1 on this small network is not much informative practically.
>
> Such basic setting in the MNIST FNN experiments allows easy reproduction of the results.
> Strong results are reported on the more challenging LSTM LM task.
>
> > Lastly, I also found a significant effort is also desired to improve the writing.
>
> We polish the paper and especially rewrite those parts after Sections 4.
>
> > The following reference also needs to be discussed in the context of using SG-MCMC in RNN. - "Scalable Bayesian Learning of Recurrent Neural Networks for Language Modeling", Zhe Gan*, Chunyuan Li*, Changyou Chen, Yunchen Pu, Qinliang Su, Lawrence Carin
>
> This work pioneers in applying SG-MCMC to Bayesian learning of RNNs, but without considering model pruning and the cost of model averaging. We have added the discussion in Related Work.

---

### Official Review · AnonReviewer3 · 2017-11-27
**A more principled way for training ensemble neural networks, nice idea and experiments**

**Rating:** 6
**Confidence:** 3

**Review:**

In this paper, the authors present a new framework for training ensemble of neural networks. The approach is based on the recent scalable MCMC methods, namely the stochastic gradient Langevin dynamics.

The paper is overall well-written and ideas are clear. The main contributions of the paper, namely using SG-MCMC methods within deep learning, and then increasing the computational efficiency by group sparsity+pruning are valuable and can have a significant impact in the domain. Besides, the proposed approach is more elegant the competing ones, while still not being theoretically justified completely.

I have the following minor comments:

1) The authors mention that retraining significantly improves the performance, even without pruning. What is the explanation for this? If there is no pruning, I would expect that all the samples would converge to the same minimum after retraining. Therefore, the reason why retraining improves the performance in all cases is not clear to me.

2) The notation |\theta_g| is confusing, the authors should use a different symbol.

3) After section 4, the language becomes quite informal sometimes, the authors should check the sentences once again.

4) The results with SGD (1 model) + GSP + PR should be added in order to have a better understanding of the improvements provided by the ensemble networks.

5) Why does the performance get worse "obviously" when the pruning is 95% and why is it not obvious when the pruning is 90%?

6) There are several typos

pg7: drew -> drawn
pg7: detail -> detailed
pg7: changing -> challenging
pg9: is strongly depend on -> depends on
pg9: two curve -> two curves

---

> ### Author Response · Authors · 2017-12-22
> **Response to Reviewer3**
>
> Thank you very much for reviewing the paper.
>
> > 1)
> As stated in the end of Section 3.2, there are two justifications for the retraining phase: First, theoretically (namely with infinite samples), model averaging does not need retraining. However, the actual number of samples used in practice is rather small for computational efficiency. So retraining essentially compensates for the limited size of samples for model averaging. Second, the MAP estimate is more likely than the network obtained just after pruning but before retraining. Retraining increases the posteriori probabilities of the networks in the ensemble and hopefully improves the prediction performance of the networks in the ensemble.
>
> Note that running SGLD enhances exploration of the model-parameter space, and we take thinned collection of samples so that there are low correlations between the samples. So in contrary to converging to the same minimum after retraining, thinned samples from SGLD would lead to neighbors of different local minima and retraining further fine-tune the paramters to take different minima.
>
> > 2)
> Thanks for your suggestion, we have changed the notation to dim(\theta_g).
>
> > 3)
> We polish the paper and especially rewrite those parts after Sections 4.
>
> > 4)
> Thanks for your suggestion. The results of SGD (1 model) + GSP + PR and SGD (ensemble) + GSP + PR have been added to Table 5, with the discussion in the paragraph before the last paragraph in Section 5.2.
> SGD (1 model)+GSP+PR can reduce the model size but the PPL is much worse than the ensemble, which clearly shows the improvement provided by the ensemble. Additionally, we compare SGLD (4 models)+GSP+PR with SGD (4 models)+GSP+PR. The two ensembles achieve close PPLs. However, SGLD ensemble learning reduces about 30% training time.
>
> > 5)
> We empirically find that 90% is the highest pruning rate without hurting performance for LSTMs.
>
> > 6)
> Typos have been fixed.

---

### Official Review · AnonReviewer2 · 2017-11-30
**A useful approach for making model averaging more feasible**

**Rating:** 6
**Confidence:** 5

**Review:**

The authors note that several recent papers have shown that bayesian model averaging is an effective and universal way to improve hold-out performance, but unfortunately are limited by increased computational costs.   Towards that end, the authors of this manuscript propose several modifications to this procedure to make it computationally feasible and indeed improve performance.

Pros:
The authors demonstrate an effective procedure for FNN and LSTMs that makes model averaging improve performance.
Empirical evidence is convincing on the utility of the approach.

Cons:
Not clear how this approach would be used with convolutional structures
Much of the benefit appears to come from the sparse prior, pruning, and retraining (Figure 3).  The model averaging seems to have a smaller contribution.  Due to that, it seems that the nature of the contribution needs to be clarified compared to the large literature on sparsifying neural networks, and the introductory comments of the paper should be rewritten to reflect that reality.

---

> ### Author Response · Authors · 2017-12-22
> **Response to Reviewer2**
>
> Thank you very much for reviewing the paper.
>
> > Not clear how this approach would be used with convolutional structures
>
> It has been shown in [1] that group Lasso regularization is effective for structured sparsity SGD learning for convolutional structures (filters, channels, filter shapes, and layer depth). It is conceivable that group Lasso used with SGLD can work for convolutional structures, by employing proper groupings like those in [1].
> [1] Wen, Wu, Wang, Chen and Li. Learning structured sparsity in deep neural networks, NIPS 2016.
>
> > The model averaging seems to have a smaller contribution.
>
> It can be seen from Figure 3(a) that as the training proceeds, more models are averaged, which consistently improves the PPLs. Also, the relationship between the performance of an ensemble and the number of models in an ensemble is examined in Figure 3(b), which clearly shows the contribution of model averaging.
>
> > Due to that, it seems that the nature of the contribution needs to be clarified compared to the large literature on sparsifying neural networks, and the introductory comments of the paper should be rewritten to reflect that reality.
>
> Literature review with regards to NN sparse structure learning and NN compression is rewritten and presented in Related Work.

---

### Public Comment · (anonymous) · 2017-11-01
**DNNs with structured sparsity learning**

In one category of your related work -- "Sparse structure learning", some previous works [1][2] used group Lasso regularization during SGD to directly learn structurally sparse DNNs for computation efficiency and memory saving. Compare or clarify the difference might make this work more comprehensive.

[1] http://papers.nips.cc/paper/6504-learning-structured-sparsity-in-deep-neural-networks.pdf
[2] http://papers.nips.cc/paper/6372-learning-the-number-of-neurons-in-deep-networks.pdf

---

> ### Author Response · Authors · 2017-12-22
> **Response**
>
> Thanks for your comment.
> As said in your comment, these previous works apply group Lasso with SGD to learn structurally sparse DNNs. They focus on point estimates and are not in the context of learning ensembles. We have added the discussion in Related Work.

---

### Author Response · Authors · 2017-12-22
**Updated paper**

Dear Reviewers,
We greatly appreciate your helpful and constructive comments on the paper. We have carefully revised the paper to incorporate your comments, adding some new results and polishing the writing for clarification. As a result, we believe that the paper has been substantially improved and strengthened.
In the following, we provide our response to your specific points.
Please find the updated paper for more details.

---

### Decision · Program_Chairs · 2018-01-29
**ICLR 2018 Conference Acceptance Decision**

**Decision:**

Reject

**Comment:**

This paper is interesting since it goes to showing the role of model averaging. The clarifications made improve the paper, but the impact of the paper is still not realised: the common confusion on the retraining can be re-examined, clarifications in the methodology and evaluation, and deeper contextulaisation of the wider literature.